METHODS AND RESOURCES

# Efficient and error-free fluorescent gene tagging in human organoids without double-strand DNA cleavage

Yannik Bollen[1,2,3☯], Joris H. Hageman[1,2☯], Petra van Leenen[1,2], Lucca L. M. Derks[2,4], Bas Ponsioen[1,2], Julian R. Buissant des Amorie[1,2], Ingrid Verlaan-Klink[1,2], Myrna van den Bos[1,2], Leon W. M. M. Terstappen[3], Ruben van Boxtel[2,4], Hugo J. G. Snippert[1,2] *

1 Molecular Cancer Research, Center for Molecular Medicine, University Medical Center Utrecht, Utrecht University, the Netherlands, 2 Oncode Institute, Utrecht, the Netherlands, 3 Medical Cell Biophysics, TechMed Centre, University of Twente, Enschede, the Netherlands, 4 Princess Máxima Center for Pediatric Oncology, Utrecht, the Netherlands

☯ These authors contributed equally to this work.
* h.j.g.snippert@umcutrecht.nl

**Data Availability Statement:** The authors confirm that all data underlying the findings are fully available without restriction. All relevant data are within the paper and its Supporting Information

## Abstract

CRISPR-associated nucleases are powerful tools for precise genome editing of model systems, including human organoids. Current methods describing fluorescent gene tagging in organoids rely on the generation of DNA double-strand breaks (DSBs) to stimulate homology-directed repair (HDR) or non-homologous end joining (NHEJ)-mediated integration of the desired knock-in. A major downside associated with DSB-mediated genome editing is the required clonal selection and expansion of candidate organoids to verify the genomic integrity of the targeted locus and to confirm the absence of off-target indels. By contrast, concurrent nicking of the genomic locus and targeting vector, known as in-trans paired nicking (ITPN), stimulates efficient HDR-mediated genome editing to generate large knock-ins without introducing DSBs. Here, we show that ITPN allows for fast, highly efficient, and indel-free fluorescent gene tagging in human normal and cancer organoids. Highlighting the ease and efficiency of ITPN, we generate triple fluorescent knock-in organoids where 3 genomic loci were simultaneously modified in a single round of targeting. In addition, we generated model systems with allele-specific readouts by differentially modifying maternal and paternal alleles in one step. ITPN using our palette of targeting vectors, publicly available from Addgene, is ideally suited for generating error-free heterozygous knock-ins in human organoids.

## Introduction

Since the development of efficient genome editing technology, molecular and cell biological research increasingly relies on genetically modified in vitro model systems. In particular, the visualization of endogenous proteins using fluorescent knock-in reporters allows for a precise assessment of their subcellular localization and dynamics during cellular homeostasis and disease [1].

files. All algorithms used for the mapping [gatk. broadinstitute.org], mutational calling [https:// github.com/ToolsVanBox/NF-IAP], and filtering of mutations [https://github.com/ToolsVanBox/ SMuRF, https://github.com/hartwigmedical/gridss-purple-linx] are publicly available. Raw FCS files are available on the FlowRepository database (flowrepository.org) and accessible using the repository ID FR-FCM-Z4PJ.

**Funding:** This work is part of the Oncode Institute, which is partly financed by the Dutch Cancer Society. HGJS received European Research Council (ERC) starting grant (IntratumoralNiche), project number 803608 (https://erc.europa.eu/ funding/starting-grants) and NWO TOP. YB was supported by a strategic alliance between University of Twente and UMC Utrecht on Advanced Biomanufacturing (to LWMMT and HJGS). The funders had no role in study design, data collection and analysis, decision to publish, or preparation of the manuscript.

**Competing interests:** The authors have declared that no competing interests exist.

**Abbreviations:** DSB, double-strand break; HDR, homology-directed repair; ITPC, in-trans paired cleavage; ITPN, in-trans paired nicking; MMEJ, microhomology-mediated end joining; NHEJ, non-homologous end joining; WGS, whole genome sequencing.

Organoids, in particular of human origin, represent next-generation model systems that recapitulate in vivo tissue architecture and functionality more accurately than 2D cell lines [2]. However, the precise engineering of large knock-in reporters in organoids can be laborious when using conventional CRISPR-mediated strategies to stimulate homology-directed repair (HDR) [3–5] or non-homologous end joining (NHEJ) [6] based editing. While generally effective, these strategies rely on the generation of genomic double-strand breaks (DSBs) by CRISPR-associated nucleases, which often result in both on- and off-target indel mutations as a consequence of error-prone repair by repeated cycles of NHEJ. On-target indels are often generated in the secondary "untargeted allele" that is not carrying the knock-in and may result in missense or nonsense mutations. In addition, while HDR generally results in error-free repair, generating knock-ins via NHEJ-based ligation of a linearized DNA fragment often results in indels within the up- and downstream junctions of the knock-in allele [6,7]. Consequently, existing knock-in protocols inherently require sequence verification of individually picked organoid clones, which is laborious, time consuming, and eliminates genetic heterogeneity in tumor-derived organoid models.

In cell lines, large knock-ins have been generated without introducing DSBs by using the partially inactivated Cas9 D10A nickase variant [8–10], which generates single-strand DNA breaks (nicks) in the genomic strand that hybridizes with the guide RNA [11]. By simultaneously nicking the genomic target locus and the extremities of both homology arms within the targeting vector, a strategy known as in-trans paired nicking (ITPN) [8], efficient knock-in alleles can be generated without double-strand DNA cleavage. Unlike conventional CRISPR/ Cas9-mediated genome editing, ITPN modifies target loci with high fidelity, since single genomic nicks are rarely mutagenic [8,12]. By avoiding double-strand DNA cleavage, ITPN enables the insertion of heterozygous reporters or pathogenic (germline) mutations with intact "untargeted" secondary alleles and with minimal risk of off-target indels. Consequently, knock-in cells can be pooled to expedite the expansion and, thus, the generation time of a knock-in line (2 weeks). Pooling successfully targeted organoids is particularly useful for organoid models where clonal selection is laborious. Furthermore, by avoiding clonal selection, preexisting genetic diversity in tumor-derived organoid lines is largely preserved. Here, we investigate the efficiency and fidelity of fluorescent gene tagging via ITPN in human organoids. In addition, we present a palette of easy-to-use targeting vector backbones and protocols for N- or C-terminal fluorescent gene tagging using ITPN.

## Results

To probe the efficiency of fluorescent knock-ins in human organoids (Fig 1A), we designed an N-terminal mScarlet knock-in at the human *SEC61B* locus. We constructed different targeting vectors in order to compare editing efficiencies of various knock-in strategies (Fig 1B). To stimulate editing via NHEJ-mediated ligation of a linearized mScarlet-coding fragment into the Cas9-generated genomic DSB [6,7], we constructed a vector carrying the mScarlet-coding sequence flanked by copies of the genomic Cas9 target site. Alternatively, we included 20 bp microhomology to stimulate genomic integration via the microhomology-mediated end joining (MMEJ) pathway [13]. In addition, we generated vectors with 1 kb homology arms following a traditional targeting vector design that is without flanking Cas9 target sites, or with flanking Cas9 target sites to support genomic integration via ITPN or in-trans paired cleavage (ITPC) [14,15].

Targeting vectors were coelectroporated with wild-type or D10A nickase SpCas9 expression constructs in a patient-derived tumor organoid model obtained from a colorectal cancer biobank [16]. We visually confirmed the expected localization of mScarlet within knock-in

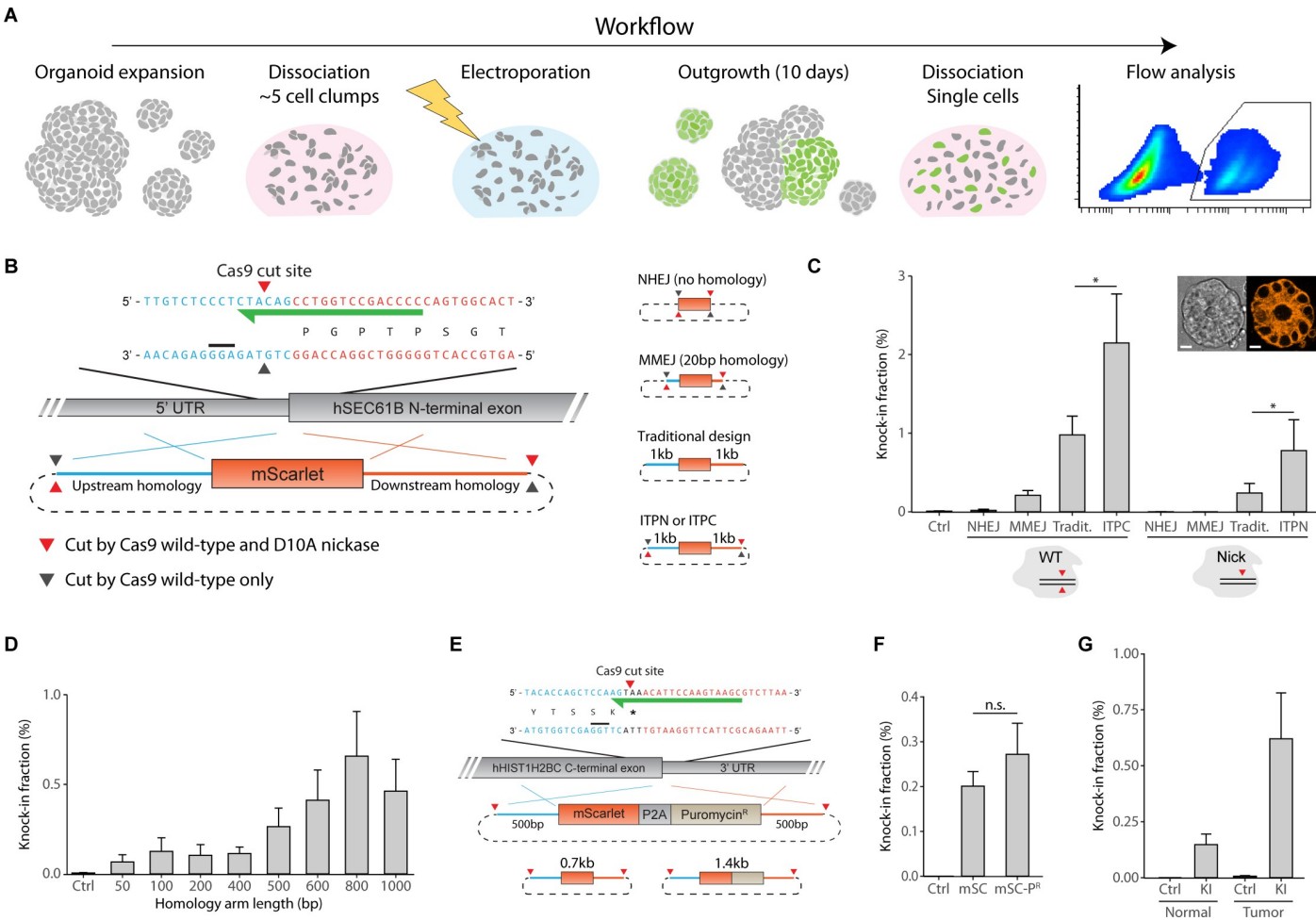

**Fig 1. Fluorescent gene tagging in human organoids without double-strand DNA cleavage.** (A) Schematic representation of the workflow used to capture fluorescent knock-in efficiencies in human organoids. To ensure optimal outgrowth post-electroporation, organoids are trypsinized to a cell suspension consisting of approximately 5 cell clumps. After electroporation, cells are allowed to expand for 10 days without selecting for cells that received the knock-in constructs. Prior to flow analysis, organoids may consist of partial knock-in populations. To capture the overall knock-in efficiency, organoid cultures are trypsinized to a single-cell suspension and flow analyzed. (B) Schematic representation of the *SEC61B* targeting strategy. mScarlet was flanked with homology arms matching up- and downstream sequences of the N-terminus of the human *SEC61B* locus, coding for Protein transport protein Sec61 subunit beta. Cas9 was targeted close to the start of the coding region using a gRNA as indicated (green arrow). Cas9 cleavage sites (triangles) and protospacer adjacent motifs (black bar) are indicated. Up- and downstream homology is represented in blue and red, respectively. Compositions of targeting vectors supporting different knock-in strategies are indicated. (C) Knock-in efficiencies of mScarlet at the human *SEC61B* locus in a patient-derived tumor organoid model using various knock-in strategies. WT or D10A nickase (Nick) SpCas9 was codelivered with targeting vectors indicated in (B). In all knock-in experiments, targeting vectors were electroporated at equimolar ratios between conditions to correct for differences in vector size. Editing efficiency (% mScarlet+ cells) was determined by single-cell flow analysis 10 days post-electroporation ($n = 3$ independent experiments). * $p < 0.05$ in a Ratio paired *t* test. Error bars indicate SEM. The inset shows representative stills of mScarlet-*SEC61B* localization in patient-derived tumor organoids (scale bar = 10 μm). (D) As in (C), knock-in efficiency of mScarlet at the human *SEC61B* locus in tumor organoids using targeting vectors with different homology arm lengths flanked by Cas9 target sites and codelivered with Cas9 D10A nickase to support ITPN ($n = 3$ independent experiments). Error bars indicate SEM. (E) As in (B), schematic showing the targeting strategy for ITPN-mediated integration of mScarlet (0.7 kb) or mScarlet-P2A-Puromycin[R] (1.4 kb) at the C-terminus of the human *HIST1H2BC* locus, coding for Histone H2B type-1C. (F) As in (C), knock-in efficiency of mScarlet (mSC; 0.7 kb) or mScarlet-P2A-Puromycin[R] (mSC-P[R]; 1.4 kb) in tumor organoids at the C-terminus of the human *HIST1H2BC* locus ($n = 3$ independent experiments). The difference between mSC and mSC-P[R] was nonsignificant in a Ratio paired *t* test. Error bars indicate SEM. (G) Knock-in efficiency of an mScarlet knock-in at the *SEC61B* locus in human colon normal and tumor organoids via ITPN using 1 kb homology arms ($n = 3$, $n = 6$ independent experiments for normal and tumor organoids, respectively). In all control conditions, the targeting vector was cotransfected with a guide targeting a different gene. The difference between normal and tumor KI organoids was nonsignificant in a two-sided unpaired *t* test. Error bars indicate SEM. Underlying data for panels C, D, F, and G are provided in S1 Data. Raw FCS files are available on the FlowRepository (FR-FCM-Z4PJ). ITPC, in-trans paired cleavage; ITPN, in-trans paired nicking; MMEJ, microhomology-mediated end joining; NHEJ, non-homologous end joining; WT, wild-type.

organoids for each condition prior to flow analysis of mScarlet$^+$ cells 10 days post-electroporation (Fig 1C). Flanking homology arms with Cas9 target sites to stimulate ITPN or ITPC resulted in substantially higher editing efficiencies when compared to a traditional targeting vector design with the same homology arm length (Figs 1C and S1A). In addition, NHEJ and MMEJ conditions underperformed when compared to targeting vectors with long homology arms, in particular when combined with nickase Cas9. Notably, ITPN resulted in a similar fraction of knock-in cells when compared to a traditional knock-in strategy that uses wild-type Cas9 and targeting vectors without flanking Cas9 target sites.

To investigate the fidelity of ITPN-mediated fluorescent knock-ins, we performed sequence analyses on polyclonal knock-in lines that were generated according to above-described conditions. To determine the risk for off-target indels, we analyzed the fidelity of the secondary allele that is not carrying the knock-in as a proxy for the likeliest candidates for off-target modifications. Using TIDE analysis [17], we show that wild-type Cas9 conditions result in a high frequency of indels within the secondary allele, whereas knock-in organoids generated via ITPN displayed >99% sequence integrity of their secondary allele (S2 Fig). Next, to investigate the fidelity of ITPN-mediated knock-ins, we generated 11 clonal knock-in lines from the ITPN condition and examined the knock-in alleles via Sanger sequencing. All knock-ins contained intact 5′ and 3′ junctions and no evidence for tandem integration was found (S3A Fig). Moreover, in agreement with previous TIDE analysis on polyclonal cultures, we confirmed the absence of indels in the untargeted allele of heterozygous clones (S3B Fig). Finally, to exclude the presence of off-target editing, we performed whole genome sequencing (WGS) on 3 out of the 11 clonal ITPN-mediated knock-in lines. We investigated the somatic mutation burden of these clones in 166 regions, which were predicted in silico to be likely off-target protospacer loci. No genomic aberrations were identified in the unmodified allele, the predicted off-target protospacer regions or the 200 bases surrounding the predicted sites (S1 Table). The lack of all variants ranging from single base substitutions to structural variation breakpoints confirms the absence of mutations due to incorrectly repaired off-target nuclease activity as well as off-target integrations of the knock-in cassette.

Collectively, these data indicate that ITPN enables highly efficient and indel-free fluorescent gene tagging in human organoids and makes sequence confirmation of clonal lines unnecessary. Consequently, all knock-in organoids can immediately be pooled to expedite the expansion of the edited organoid line and to maintain genetic diversity of patient-derived tumor organoid models.

Traditional design of targeting vectors requires long homology arms to maximize the chance of homologous recombination between the genomic locus and targeting vector. However, vectors with long homology arms are challenging to assemble and are inconvenient for locus-specific genotyping by PCR. To investigate whether efficiency of fluorescent gene tagging is lost when ITPN is mediated by shorter homology arms, we generated a series of targeting vectors with decreasing homology. At the *SEC61B* locus, the homology demand of ITPN-mediated mScarlet integration peaked at 800 bp (Figs 1D and S1B). While vectors with shorter homology arms were accompanied with lower editing efficiencies, they were sufficient to generate knock-in lines and may be preferred in situations of challenging vector assembly and/or genotyping.

Next, to probe whether knock-in size influences editing efficiency, we designed a C-terminal knock-in at the *HIST1H2BC* locus and constructed targeting vectors with 500 bp homology to integrate either mScarlet (0.7 kb) or mScarlet-P2A-Puro (1.4 kb) (Fig 1E). Surprisingly, we found no substantial difference in knock-in efficiency between the 2 targeting vector variants, suggesting that a knock-in size in the range of <1.4 kb has no notable influence on editing efficiency via ITPN (Figs 1F and S1C).

Since *SEC61B* and *HIST1H2BC* are ubiquitously expressed genes, we decided to investigate if we could knock-in mScarlet-P2A-Blast into normal human colon organoids at the C-terminus of *KRT20*, which is exclusively expressed in differentiated cells. Following a short pulse of Blasticidin selection, we observed clonal organoids with a subpopulation of cells showing the expected cytoplasmic red fluorescence (S4 Fig). Since differentiated cells do not form organoids as efficiently as stem cells, lines that involve fluorescent knock-ins in differentiation genes such as *KRT20* are best generated either using a short pulse of selection or by manually picking and pooling clonal organoids that contain (some) fluorescent cells.

Finally, we compared the efficiency of an N-terminal mScarlet knock-in at the *SEC61B* locus between tumor and normal colon organoids (Figs 1G and S1D). The knock-in efficiency in tumor organoids was higher (although not significant), which may be attributed to a difference in culture conditions and electroporation efficiency.

A major downside of generating targeting vectors with homology arms flanked by Cas9 target sites at their extremities is the time-intensive molecular cloning. To expedite the cloning of targeting vectors for fluorescent gene tagging at either the N- or C-terminus, we generated a series of minimalistic targeting vector backbones allowing seamless one-step integration of both homology arms using SapI-based Golden Gate assembly [4] (Fig 2A). Targeting vector backbones carrying state-of-the-art monomeric fluorescent proteins are made available from Addgene, including optional P2A-linked selection elements (Fig 2B). Using our optimized vector backbones, targeting vectors can be assembled in the same amount of cloning time as is required for the insertion of gRNA oligos into Cas9 expression vectors. Consequently, when using our vector backbones for ITPN, fluorescent reporter alleles in cell lines and organoid

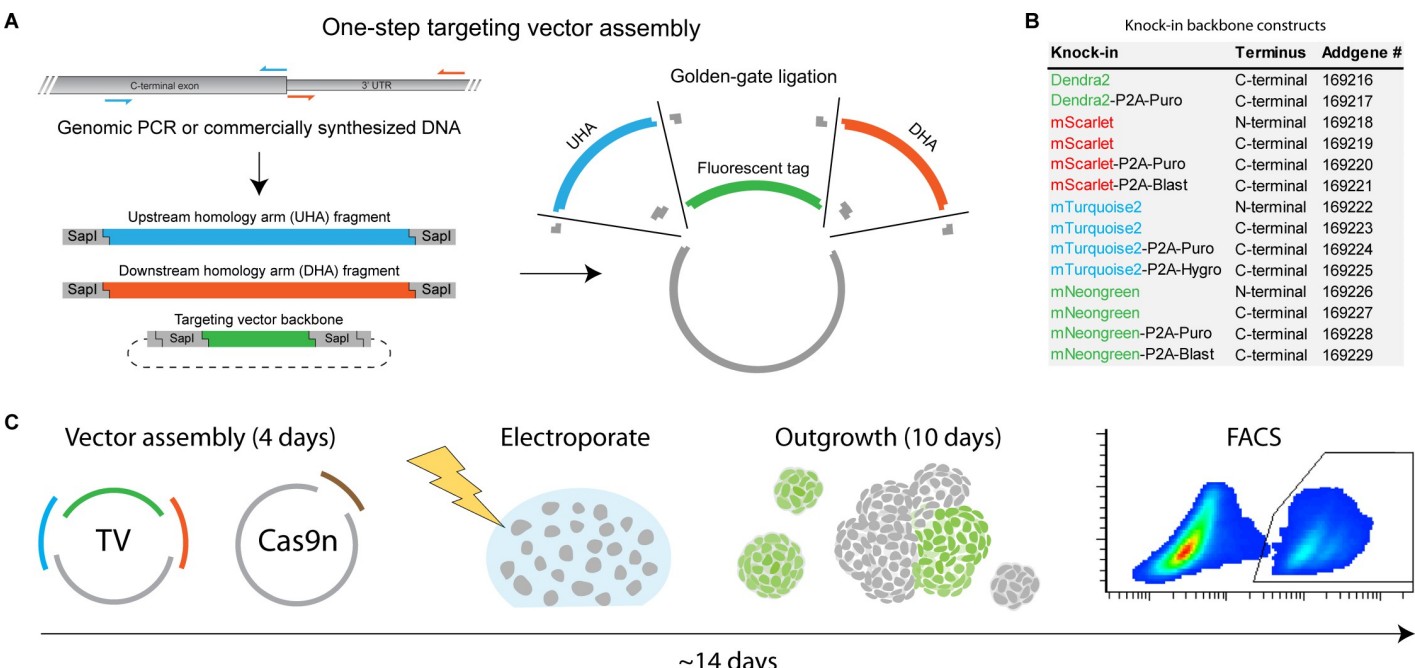

**Fig 2. One-step targeting vector assembly and ITPN expedite fluorescent gene tagging.** (A) Schematic outline of one-step TV assembly via SapI-based golden gate-mediated homology arm ligation. Homology arms can be amplified from genomic DNA or ordered as commercially synthesized DNA fragments. (B) Overview of knock-in backbone constructs available from Addgene. Knock-in backbones contain one of 4 different fluorescent proteins and optional P2A-linked resistance cassettes. Backbone constructs are suitable for knock-ins at either the C- or N-terminus, as indicated. (C) Schematic workflow outlining fluorescent gene tagging in organoids using ITPN. Following electroporation, organoids generally require approximately 10 days of outgrowth before FACS purification of fluorescent knock-in cells. Alternatively, fluorescent clonal organoids can be handpicked and pooled. Sequence verification of individually picked clonal lines is not required when editing via ITPN. ITPN, in-trans paired nicking; TV, targeting vector.

models can be generated in as little as 2 weeks, including molecular cloning procedures for vector assembly (Fig 2C). We summarized our recommendations for knock-in design and one-step targeting vector assembly in a protocol (S1 File). In addition, new variants of targeting vector backbones, e.g., replacing the donor with a different fluorescent protein sequence, can be generated in a short amount of time (S2 File).

To probe the efficiency of ITPN using our newly designed targeting vectors, we generated triple fluorescent knock-ins by simultaneous targeting of 3 separate genomic loci. Specifically, we targeted the C-terminus of the *HIST1H2BC* locus to knock-in mTurquoise2-P2A-puromycin^R, the C-terminus of the *CDH1* locus to knock-in mScarlet, and as a third locus, we included an N-terminal knock-in of mNeongreen at either the *LMNA*, *SEC61B*, or *MAP4* locus (Fig 3A). DNA cocktails containing different combinations of targeting vectors and their respective Cas9 expression constructs were electroporated into fractionated tumor organoids. Organoids were allowed to form for 10 days without puromycin selection prior to quantification of the raw knock-in efficiencies by single-cell flow analysis. As expected, in all 3 conditions, the knock-in fractions were dominated by cells that carried single knock-ins in either one of the targeted genes. However, we readily detected cells carrying multiple knock-ins, including cells where all 3 genes were edited simultaneously (Fig 3B). The overall knock-in efficiencies for each targeted gene and the fraction of cells carrying multiple knock-ins are summarized in Fig 3C. To confirm the fidelity of the gene fusions, we generated polyclonal triple knock-in lines from each editing condition by manual picking and pooling clonal triple positive organoids. TIDE analysis again confirmed the absence of on-target indels in the untargeted alleles of all edited genes (S5 Fig). In addition, we confirmed the intended integration of the knock-in via Sanger sequencing (S6 Fig). Next, we recorded overnight growth of our TKI-3 knock-in line using live-cell imaging to demonstrate normal growth behavior and phenotype (Fig 3D). Each channel could be recorded without excessive bleaching, allowing a multidimensional dynamic readout of chromosomal instability during mitosis, including chromatin errors (H2B1C), spindle assembly (MAP4), and membrane defects or binucleation (CDH1).

Taken together, these results demonstrate that ITPN maintains high levels of fidelity across different genomic loci and allows multiplexed fluorescent gene tagging in human organoids. Using conventional editing protocols, generating organoid lines carrying multiple fluorescent knock-ins is highly laborious. By using ITPN, organoids with multiple edits can be generated within 2 weeks. Alternatively, in case an attempt to multiplex gene targeting fails, cells with a single knock-in can be pooled and retargeted. Moreover, we generated the same combinations of triple knock-ins in 2 rounds of targeting and used intermediate antibiotic selection to enrich for knock-in cells instead of manual picking (S7 Fig).

Since the sequence integrity of the untargeted allele that is not carrying the knock-in is maintained when editing via ITPN, this secondary allele can be retargeted using the same locus-specific targeting vector to obtain homozygous knock-ins. This also enables straightforward differential modification of maternal and paternal alleles by offering 2 different targeting vectors for the same locus. To investigate if ITPN allows the simultaneous generation of biallelic knock-ins carrying different fluorescent tags within each allele, we targeted the *SEC61B*, *MAP4*, and *HIST1H2BC* loci in tumor organoids with both mNeongreen and mScarlet targeting vectors. Flow analysis at 10 days post-electroporation confirmed the presence of a double-positive cell population for each targeted locus (Fig 4A). Genotyping of manually picked lines confirmed correct modification of each allele (S8 Fig). In addition, imaging of biallelic knock-in organoids confirmed the detection of both allele-specific reporters (Fig 4B). Next, we performed live-cell imaging of our *HIST1H2BC* double knock-in organoids and assessed the biallelic fluorescent output (green versus red) for single cells over time (Fig 4C), as a straightforward showcase how differential allele-specific modifications could be used to study allele-

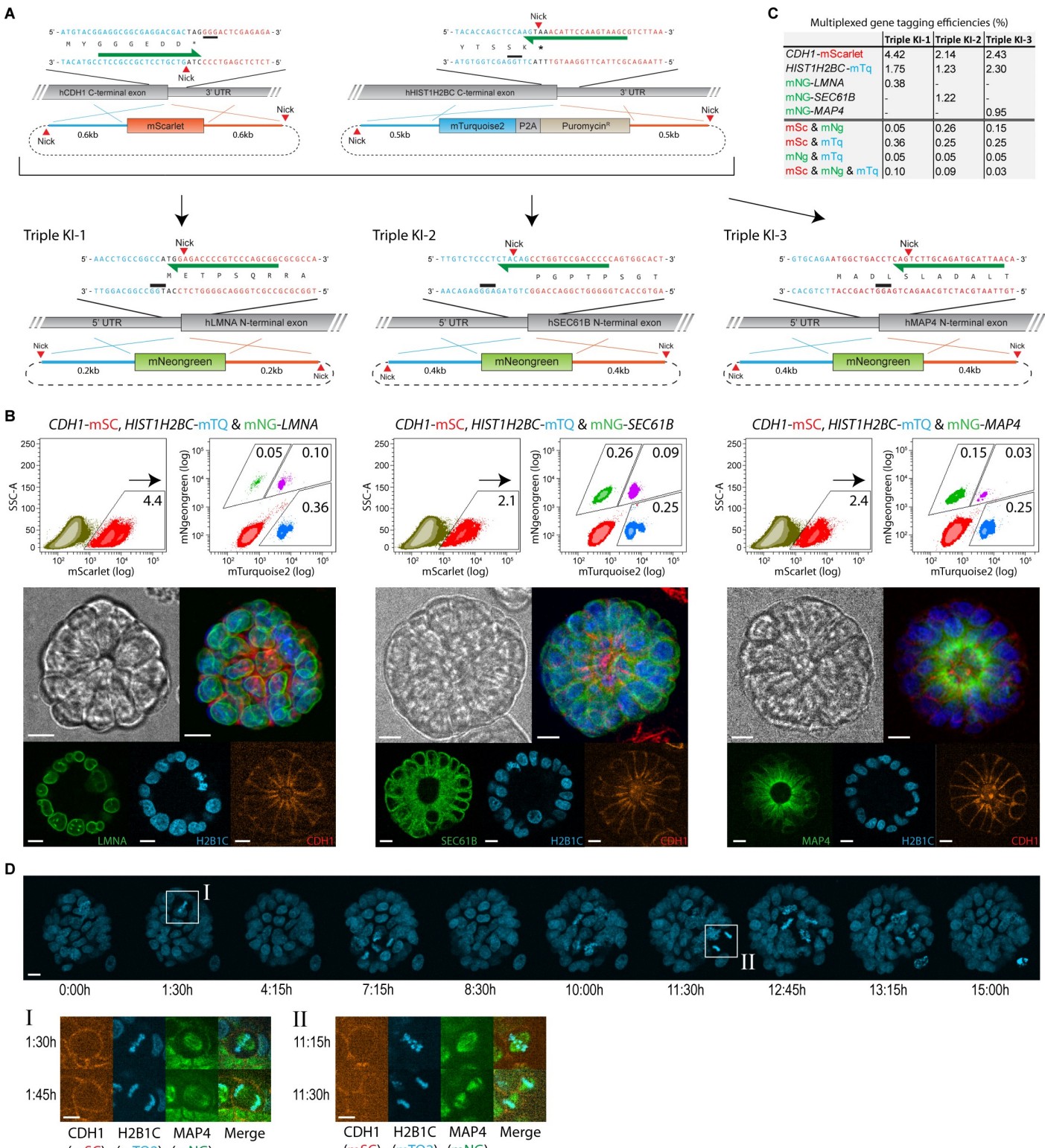

**Fig 3. Multiplexed fluorescent gene tagging in human organoids using ITPN.** (A) Multiplexed fluorescent gene tagging in tumor human colon organoids at 3 different genomic loci using ITPN. C-terminal integrations of mScarlet at the *CDH1* locus and mTurquoise2-P2A-Puromycin into the *HIST1H2BC* locus were combined with N-terminal integration of mNeongreen at either the *LMNA*, *SEC61B*, or *MAP4* locus. In the schematics: Cas9 D10A nick positions (red triangles) and protospacer adjacent motifs (black bars) are indicated for each knock-in design, as well as the gRNA used (green arrow). Organoids were electroporated simultaneously with all 3 targeting vectors to generate one-step multiplexed triple knock-ins. (B) All 3 targeting combinations yielded triple knock-in populations

with practical efficiencies, as indicated by flow analysis (numbers indicate frequencies (%) of knock-in cells within the entire targeted cell population). Imaging snapshots show the expected subcellular localization of each fusion protein (scale bar = 10 μm). Raw FCS files are available on the FlowRepository (FR-FCM-Z4PJ). (C) Overview of the multiplexed gene tagging efficiencies as determined by flow cytometry analysis. Raw FCS files are available on the FlowRepository (FR-FCM-Z4PJ). (D) Live-cell imaging of tumor human colon organoids carrying *CDH1-mScarlet*, *HIST1H2BC-mTurquoise2*, and *mNeongreen-MAP4* knock-ins. The top panel shows representative stills of organoid growth over time (scale bar = 10 μm). For divisions I and II, snapshots of each channel are shown in metaphase and anaphase (scale bar = 5 μm). ITPN, in-trans paired nicking.

specific expression levels [18,19]. This proof of principle underscores the power of ITPN to create allele-specific readouts that, depending on the knock-in template, can be applied to address many biological questions, ranging from allele-specific expression patterns to differential biochemical properties between wild-type and mutant proteins.

## Discussion

Here, we show that large knock-ins such as fluorescent gene tags can be generated in human organoids with high efficiency and fidelity using ITPN. The strategy is superior to conventional Cas9-mediated genome editing as ITPN is DSB independent, which minimizes the risk

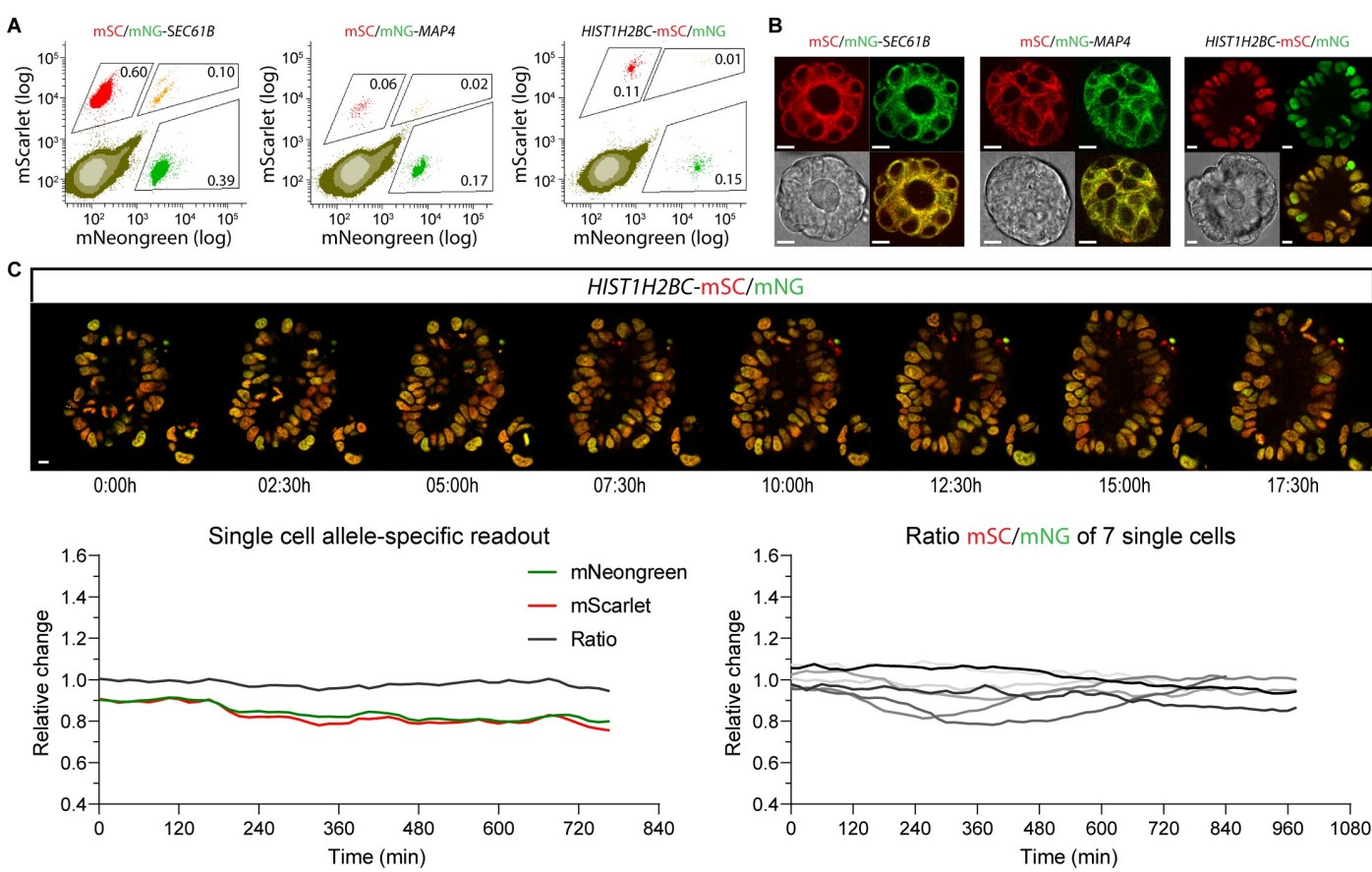

**Fig 4. One-step differential fluorescent knock-ins at a single locus.** (A) The *SEC61B*, *MAP4*, and *HIST1H2BC* loci of a patient-derived tumor organoid model were targeted via ITPN with both mNeongreen and mScarlet targeting vectors according to the design shown in Fig 3A. Flow analysis was performed 10 days post-electroporation. The percentage of cells carrying a single knock-in and cells carrying both knock-ins are indicated. Raw FCS files are available on the FlowRepository (FR-FCM-Z4PJ). (B) Representative stills of tumor organoids carrying biallelic mNeongreen and mScarlet modifications at either the *SEC61B*, *MAP4*, and *HIST1H2BC* loci (scale bar = 10 μm). (C) Allele-specific readout in human colorectal cancer organoids containing differentially tagged *HIST1H2BC* alleles. Organoids carrying mNeongreen and mScarlet knock-ins at the *HIST1H2BC* locus were live-cell imaged for 18 hours. Top panels show imaging stills of the green/red composite over time. The left bottom graph shows the changes of allele-specific output over time for a single representative cell. The corrected fluorescent signals of mScarlet, mNeongreen, as well as the ratio mScarlet/mNeongreen are plotted. The right bottom graph shows 7 out of 17 analyzed single cell ratios (mScarlet/mNeongreen). Underlying data are provided in S1 Data. ITPN, in-trans paired nicking.

of undesired mutations, both at potential off-target sites as well as within the untargeted allele that does not carry the knock-in. This also implies that ITPN is ideally suited for the introduction of heterozygous pathogenic (germline) mutations. An additional advantage is that sequence verification of individually picked clonal lines becomes obsolete and thus that all de novo generated knock-in cells can immediately be pooled. This accelerates expansion of the early culture and reduces the overall generation time of knock-in models. In addition, pooling all de novo generated knock-in cells preserves the genetic heterogeneity present in the original culture, which is important when working with patient-derived tumor organoid models. To circumvent labor-intensive molecular cloning of targeting vectors, we generated a palette of vector backbones that can be locus customized in the same amount of cloning time required for gRNA oligo insertion into Cas9 expression vectors. Moreover, our targeting vector backbone design is modular, so that the donor template itself can easily be adapted to more complex and sophisticated reporter designs [5]. Our vector backbones are available on Addgene, and our protocols (S1 and S2 Files) contain detailed instructions for their application.

While absolute knock-in efficiency depends on many variables, the most practical determinant of a successful strategy is that knock-ins are consistently obtained with each attempt. To showcase the robust efficiency of ITPN in organoids, we generated triple knock-ins at 3 independent genomic loci in a single targeting round, and we accomplished a one-step generation of biallelic knock-ins carrying allele-specific reporters. In our hands, efficiency of ITPN is superior to CRISPR strategies that use conventional donor templates. In agreement with earlier reports [8,9,14,15], an important variable that influences overall efficiency was the ITPN or ITPC of the donor. Although NHEJ-mediated knock-ins by strategies such as CRISPR-HOT [6] minimize the need for molecular cloning, our results indicate that ITPN substantially improves efficiency and fidelity when compared to NHEJ-mediated strategies, while our dedicated vector backbones minimize molecular cloning to a similar extent.

Due to the near exact sequence similarity between maternal and paternal alleles, differences in expression patterns between 2 alleles of a given gene are poorly understood. Likewise, changes in subcellular localization, protein interactions, and/or biochemical properties between wild-type and mutant proteins, such as oncogenes, are rarely examined in the same cells due to the near impossible task of modifying both alleles independently with different tags. Since ITPN generates knock-in alleles without modifying the untargeted allele, the same locus can be retargeted with an alternative donor template. As a proof of principle, we generated a biallelic knock-in with different fluorophores at the *HIST1H2BC*, *SEC61B*, and *MAP4* loci in a single targeting round. Accurate estimation of allelic imbalance is important to understand genetic and epigenetic mechanisms of gene regulation, and dysregulation during carcinogenesis.

In conclusion, ITPN is a versatile strategy that generates fast and efficient knock-ins in human organoids. We envision that our approach can easily be applied to organoid models derived from other tissues or sources, such as pluripotent stem cell-derived organoid models. Various CRISPR-mediated knock-in strategies have been reported to date that reach sufficient efficiencies to make genetic editing practical in organoid models. ITPN is comparable in terms of efficiency but stands out as being DSB independent and therefore has the highest intrinsic fidelity of precise genome editing. In combination with the seamless one-step generation of targeting vectors, ITPN represents an important technological advance in generating high-fidelity knock-in model systems.

## Materials and methods

### Vector assembly

Targeting vector backbones were generated by recombinase-based seamless assembly (In-Fusion cloning, Takara Biotech) of a commercially synthesized DNA fragment (IDT or

Genscript) carrying mNeongreen [20], mTurquiose2 [21], mScarlet [22], or photoconvertible Dendra2 [23] with optional P2A-selection elements (S2 Table) into a PCR-amplified generic backbone fragment (FWD: tcctcgctcactgactcgct, REV: gcggtattttctccttacgcatctg). See S2 File for a detailed explanation of how to generate new targeting vector backbones. To generate locus-specific targeting vectors, commercially synthesized homology arm fragments (S3 Table) were inserted into targeting vector backbones using SapI-based golden gate assembly as previously described [4]; see S1 File for a more detailed protocol. Cas9 wild-type (addgene #48139) and Cas9 D10A nickase (addgene #48141) locus-specific expression vectors were generated according to published protocols [24].

## Organoid culture

Patient-derived tumor organoid with identifier P9T (PDTO-9) was obtained from a previously published colorectal cancer biobank [16]. PDTO-9 was maintained at 37˚C with 5% $CO_2$ atmosphere seeded in RGF Basement Membrane Extract (BME), Type 2 (Cultrex). Culture media consisted of advanced DMEM/F12 (Gibco) supplemented with penicillin–streptomycin (Lonza, 10 U ml$^{-1}$), GlutaMAX (Gibco, 1x), HEPES buffer (Gibco, 10 mM), Noggin-conditioned medium (10%), R-spondin1-conditioned medium (10%), B-27 (Gibco, 1x), nicotinamide (Sigma-Aldrich, 10 mM), *N*-acetylcysteine (Sigma-Aldrich, 1.25 mM), SB202190 (Gentaur, 10 μM), A83-01 (Tocris, 500 nM), and recombinant human EGF (PeproTech, 50 ng ml$^{-1}$). PDTO-9 cultures were passaged weekly and maintained below passage 10. Briefly, PDTOs were dissociated using trypsin–EDTA (Sigma-Aldrich) and seeded in BME in a prewarmed 24-well plate. ROCK inhibitor Y-27632 (Gentaur, 10 μM) was added to culture medium upon plating for 2 days.

Normal human colon organoids were maintained at 37˚C with 5% $CO_2$ atmosphere seeded in growth factor reduced Matrigel (BD Biosciences). Culture media [25] consisted of advanced DMEM/F12 (Gibco) supplemented with penicillin–streptomycin (Lonza, 10 U ml$^{-1}$), Gluta-MAX (Gibco, 1x), HEPES buffer (Gibco, 10 mM), Noggin-conditioned medium (10%), R-spondin1-conditioned medium (20%), B-27 (Gibco, 1x), *N*-acetylcysteine (Sigma-Aldrich, 1.25 mM), A83-01 (Tocris, 500 nM), recombinant human EGF (PeproTech, 50 ng ml$^{-1}$), recombinant human IGF-1 (BioLegend, 100 ng/ml), recombinant human FGF-basic (Pepro-tech, FGF-2 50 ng/ml), and 0.5 nM Wnt surrogate (U-protein Express). Organoid were passaged as described above.

## Organoid electroporation

To generate knock-ins in either normal or colorectal tumor organoids, $1 \times 10^6$ cells at approximately 5 cell clumps were coelectroporated with 15 μg DNA, at a 1:1 ratio of Cas9 (Addgene #48139) or Cas9 D10A nickase (Addgene #48141) and targeting vector using the NEPA21 Super Electroporator (Nepagene) following described conditions [26].

## Flow analysis

Organoids were trypsinized and filtered through a CellTrics 10 μm sieve (Sysmex) to obtain a single-cell suspension. To quantify the knock-in efficiency, cells were flow analyzed (FACSCelesta, BD) at least 10 days post-electroporation. Gates were set based on a negative control/ population.

## Genotyping and TIDE analysis

Polyclonal or clonal knock-in cultures were established via manual picking or FACS. Site-specific integrations were confirmed by genotyping PCRs on genomic DNA extract using locus-

specific primer sets (S4 Table), followed by Sanger sequencing. TIDE analysis was performed on Sanger sequencing data of secondary "non-knock-in" alleles using the Sanger sequencing read of the parental organoid line as a control sample chromatogram.

## Whole genome sequencing and read mapping

The genomic DNA of 3 clonally expanded ITPN-mediated *SEC61B*-mScarlet organoid lines was isolated using the QIAamp DNA Micro Kit according to the manufacturer's instructions. Illumina sequencing libraries were generated using 200 ng of genomic DNA according to standard protocols (Illumina). Following WGS to a base coverage of 15x (Illumina NovaSeq 6000, $2 \times 150$ bp), initial processing of the sequence reads was performed using the complete analysis pipeline available at https://github.com/UMCUGenetics/NF-IAP. Briefly, the Burrows-Wheeler Aligner v0.7.17 mapping tool was used to map sequence reads against human reference genome GRCh38 with settings "bwa mem -c 100 –M" [27]. Next, duplicate reads were flagged with Sambamba v0.6.8 and the Genome Analysis Toolkit (GATK) v4.1.3.0 was used for realignment [28].

## Variant calling and filtering

Next, variants were multisample called with the GATK HaplotypeCaller v4.1.3.0 and GATK-Queue v.4.1.3.0, based on default settings and the additional option "EMIT_ALL_CONFIDENT_SITES." Subsequently, GATK VariantFiltration v4.1.3.0 was used to evaluate the quality of the variant positions, with options -snpFilterName SNP_LowQualityDepth -snpFilterExpression "QD < 2.0" -snpFilterName SNP_MappingQuality -snpFilterExpression "MQ < 40.0" -snpFilterName SNP_StrandBias -snpFilterExpression "FS > 60.0" -snpFilterName SNP_HaplotypeScoreHigh -snpFilterExpression "HaplotypeScore > 13.0" -snpFilterName SNP_MQRankSumLow -snpFilterExpression "MQRankSum < −12.5" -snpFilterName SNP_ReadPosRankSumLow -snpFilterExpression "ReadPosRankSum < −8.0" -snpFilterName SNP_HardToValidate -snpFilterExpression "MQ0 > = 4 && ((MQ0 / (1.0 * DP)) > 0.1)" -snpFilterName SNP_LowCoverage -snpFilterExpression "DP < 5" -snpFilterName SNP_VeryLowQual -snpFilterExpression "QUAL < 30" -snpFilterName SNP_LowQual -snpFilterExpression "QUAL > = 30.0 && QUAL < 50.0" -snpFilterName SNP_SOR -snpFilterExpression "SOR > 4.0" -cluster 3 -window 10 -indelType INDEL -indelType MIXED -indelFilterName INDEL_LowQualityDepth -indelFilterExpression "QD < 2.0" -indelFilterName INDEL_StrandBias -indelFilterExpression "FS > 200.0" -indelFilterName INDEL_ReadPosRankSumLow -indelFilterExpression "ReadPosRankSum < −20.0" -indelFilterName INDEL_HardToValidate -indelFilterExpression "MQ0 > = 4 && ((MQ0 / (1.0 * DP)) > 0.1)" -indelFilterName INDEL_LowCoverage -indelFilterExpression "DP < 5" -indelFilterName INDEL_VeryLowQual -indelFilterExpression "QUAL < 30.0" -indelFilterName INDEL_LowQual -indelFilterExpression "QUAL > = 30.0 && QUAL < 50.0" -indelFilterName INDEL_SOR -indelFilterExpression "SOR > 10.0."

Low-quality and subclonal mutations accumulated during clonal expansion of the organoid lines were excluded by annotating using SMuRF release 2.1.2 as described [29] (https://github.com/ToolsVanBox/SMuRF). We included all variants in each clone at autosomal or X chromosomes present in less than 3 clonal samples that passed VariantFiltration, with a GATK phred-scaled quality score ≥60; minimum base coverage of 5X, a mapping quality ≥30, and a variant allele frequency of at least 0.15 [29,30]. Structural variation calling was performed with the GRIDSS-purple-linx pipeline v1.3.2, using all paired combinations of the 3 WGS samples as tumor-normal pairs [31].

### Analysis of in silico predicted off-target regions

All potential off-target protospacer regions for the guide sequence 5′-GGGGTCGGAC-CAGGCTGTAG-3′ were predicted using the publicly available CasOFFinder tool [32], using an NGG PAM and allowing up to 4 mismatches. As regions of interest, both the potential off-target protospacer regions as well as the flanking 200 bases were considered. Using BEDtools v2.27.1, all variants that passed filtering by SMuRF were intersected with the regions of interest [33]. In addition, all start and end coordinates of the structural variations called by GRIDSS-purple-linx were intersected with the same potential off-target genomic regions.

### Live organoid imaging

To support live-cell microscopy of organoids, PDTO-9 or normal organoid cultures were passaged 5 to 7 days prior to imaging. PDTOs were harvested 24 hours before imaging and resuspended in an ice-cold mix of culture media containing 50% v/v BME or 90% v/v Matrigel. The organoid suspension was then seeded in an ice-cold glass bottom *WillCo*-dish (WillCo Wells B.V.) coated with a thin film of BME or Matrigel. Organoids were allowed to settle on ice before gel polymerization at 37˚C and addition of culture media. Outgrowth was captured overnight on a spinning disk confocal system (Nikon, 15-minute frame rate, z-step 1.4 μm). Imaging data were analyzed with Fiji (ImageJ). For the *HIST1H2BC* biallelic knock-in organoids, a custom-made analysis macro [34] was used to track single cells and monitor their mNeongreen and mScarlet signals. Bleach correction (per channel, per time point) was performed based on integral fluorescence signals of the corresponding organoids.

### Supporting information

**S1 Fig. Flow cytometry analyses for the measurement of knock-in efficiencies.** Representative flow cytometry plots for the analysis of knock-in efficiencies including nonedited controls are shown. Panel (A), (B), (C), and (D) are related to Fig 1C, 1D, 1F, and 1G, respectively. Raw FCS files are available on the FlowRepository (FR-FCM-Z4PJ). ITPC, in-trans paired cleavage; ITPN, in-trans paired nicking; MMEJ, microhomology-mediated end joining; NHEJ, nonhomologous end joining; wt, wild-type.
(EPS)

**S2 Fig. On-target TIDE analysis of different knock-in strategies at the *SEC61B* locus.** To perform TIDE analyses, genomic DNA was extracted from polyclonal knock-in populations generated either via MMEJ, traditional targeting, ITPC (with wild-type Cas9), or ITPN (with Cas9 D10A nickase) (see Fig 1C). Polyclonal lines were generated by manually pooling approximately 10 clonal knock-in organoids. TIDE analysis was performed using the Sanger sequencing read of the parental patient-derived tumor organoid model as a control sample chromatogram. The NHEJ condition codelivered with wild-type Cas9 was omitted from this analysis as there were too few knock-in organoids available to generate a polyclonal line. The percentage of reads containing indels is displayed in the left upper corners. Underlying data are provided in S2 Data. ITPC, in-trans paired cleavage; ITPN, in-trans paired nicking; MMEJ, microhomology-mediated end joining; NHEJ, nonhomologous end joining; WT, wild-type.
(EPS)

**S3 Fig. Fidelity of ITPN-mediated fluorescent knock-ins at the *SEC61B* locus.** Sanger sequencing was performed on the *SEC61B* locus of 11 clonal patient-derived tumor organoid knock-in lines generated via ITPN (see Fig 1C). Knock-in lines were generated by handpicking individual large clonal knock-in organoids. The 5′ and 3′ junctions of the knock-in allele and the target region of the "untargeted" allele is shown for each clone. Clone no.7 carries a

homozygous mScarlet knock-in at the *SEC61B* locus and therefore does not contain an untargeted allele. Underlying data are provided in S2 Data. ITPN, in-trans paired nicking.
(TIF)

**S4 Fig. KRT20-mScarlet knock-in in normal human colon organoids.** (A) Schematic representation of the *KRT20* targeting strategy to knock in mScarlet-P2A-Blasticidin[R]. The donor was flanked with homology arms matching up- and downstream sequences of the C-terminus of the human *KRT20* locus. Cas9 D10A nickase was targeted close to the stop codon using a gRNA as indicated (green arrow). Cas9 D10A nickase cleavage sites (triangles) and protospacer adjacent motifs (black bar) are indicated. Up- and downstream homology is represented in blue and red, respectively. Knock-in organoids were generated by electroporating normal human colon organoid clumps followed by outgrowth for 10 days. To select for successfully targeted cells, organoids were treated for 7 days with Blasticidin, followed by manual picking of a clonal organoid containing red fluorescent cells. (B) Fluorescent images of representative *KRT20*-mScarlet-P2A-Blasticidin[R] knock-in organoids containing both KRT20-positive and KRT20-negative cells (red: KRT20-mScarlet). Scale bar = 50 μm. (C) Sanger sequencing was performed on the *KRT20*-mScarlet-P2A-Blasticidin[R] knock-in line. The 5′ and 3′ junctions of the knock-in allele are intact. Underlying data are provided in S2 Data.
(EPS)

**S5 Fig. TIDE analysis of multiplexed triple knock-in lines in tumor human colon organoids.** TIDE analysis of the "untargeted" alleles for all targeted genes in the one-step triple knock-in lines (see Fig 3). TIDE was performed using the Sanger sequencing read of the parental tumor colon organoid model as a control sample chromatogram. The SEC61B allele of Triple KI-2 was not analyzed because it failed quality control. Underlying data are provided in S2 Data.
(EPS)

**S6 Fig. Sanger sequencing of 5′ and 3′ knock-in allele junctions in triple knock-in tumor colon organoids.** Sanger sequencing was performed on polyclonal triple knock-in tumor colon lines generated via manual picking of triple positive clonal organoids. The 5′ and 3′ junctions of each knock-in allele are shown for all 3 multiplexed knock-in conditions (see Fig 3). Underlying data are provided in S2 Data.
(TIF)

**S7 Fig. Two-step triple knock-ins in human colon cancer organoids using antibiotic enrichment.** (A) Schematic representation showing the generation of triple fluorescent knock-ins in human colon cancer organoids in 2 rounds of gene targeting. mScarlet-Blast[R] was integrated into the *CDH1* locus, coding for E-cadherin, followed by Blasticidin selection for 2 weeks to enrich for knock-in organoids. Subsequent handpicking generated a pure line (99.5%, Scarlet+). In a second round of gene targeting, mTurquoise2-Puro[R] was integrated into the *HIST1H2BC* locus combined with mNeongreen integrations into either the LMNA, SEC61B, or MAP4 loci. In the schematics: Cas9 (D10A) nick positions (red triangles) and protospacer adjacent motifs (black bar) are indicated for each knock-in design, including gRNA used (green arrow). (B) FACS plots of triple knock-in organoids after the second targeting round and puromycin selection for 10 days. Puromycin enrichment resulted in a near complete selection for HIST1H2B-mTQ2-Puro[R] positive cells (purple and blue populations combined). Numbers in FACS plots indicate frequencies (%) of knock-in populations within the entire targeted cell population. Raw FCS files are available on the FlowRepository (FR-FCM-Z4PJ).
(EPS)

**S8 Fig. Sanger sequencing of differential biallelic modifications in a patient-derived tumor organoid line.** Sanger sequencing was performed on polyclonal biallelic tumor knock-in lines carrying both mScarlet and mNeongreen integrations at either the *SEC61B*, *MAP4*, or *HIST1H2BC* locus. The 5′ and 3′ junctions of each knock-in allele are shown for all 3 biallelic knock-in conditions (see Fig 4A). For *HIST1H2BC*-mSC, the endogenous stop codon of the *HIST1H2BC* locus was maintained.
(TIF)

**S1 Table. Presence of off-target effects (small indels and structural variations breakpoints) in in silico predicted off-target regions (spacer and flanking regions) in WGS of 3 clonal ITPN-mediated *SEC61B*-mScarlet knock-in organoid lines.** ITPN, in-trans paired nicking; WGS, whole genome sequencing.
(XLSX)

**S2 Table. Targeting vector backbone fragments.**
(XLSX)

**S3 Table. Targeting vector homology arm sequences.**
(XLSX)

**S4 Table. Locus-specific primer sets for genotyping and TIDE.**
(XLSX)

**S1 Data. Data underlying Figs 1C, 1D, 1F, 1G, and 4C.** (A-D) Knock-in efficiencies (% of total cell population) as determined by flow cytometry analysis for Fig 1C, 1D, 1F and 1G. (E) Fluorescent signals reported in Fig 4C. Values are corrected as explained in the methods.
(XLSX)

**S2 Data. Raw Sanger and TIDE sequencing results for S2–S6 and S8 Figs.**
(ZIP)

**S1 File. Design considerations and cloning protocol for the one-step targeting vector assembly to obtain fluorescent gene tagging using in-trans paired nicking or cleavage.**
(DOCX)

**S2 File. Personalize your one-step targeting vector backbone by replacing the current donor templates.**
(DOCX)

## Acknowledgments

We thank members of the Snippert laboratory for reagents, suggestions, and discussions. We thank Markus J. van Roosmalen for advice on the WGS analysis.

## Author Contributions

**Conceptualization:** Yannik Bollen, Hugo J. G. Snippert.

**Formal analysis:** Yannik Bollen, Joris H. Hageman, Petra van Leenen, Lucca L. M. Derks, Bas Ponsioen, Ruben van Boxtel, Hugo J. G. Snippert.

**Funding acquisition:** Leon W. M. M. Terstappen, Hugo J. G. Snippert.

**Investigation:** Yannik Bollen, Joris H. Hageman, Petra van Leenen, Lucca L. M. Derks, Ingrid Verlaan-Klink, Myrna van den Bos.

**Methodology:** Yannik Bollen, Joris H. Hageman, Julian R. Buissant des Amorie, Hugo J. G. Snippert.

**Resources:** Joris H. Hageman, Leon W. M. M. Terstappen, Hugo J. G. Snippert.

**Supervision:** Leon W. M. M. Terstappen, Ruben van Boxtel, Hugo J. G. Snippert.

**Visualization:** Yannik Bollen, Joris H. Hageman, Petra van Leenen, Lucca L. M. Derks, Bas Ponsioen, Julian R. Buissant des Amorie, Hugo J. G. Snippert.

**Writing – original draft:** Yannik Bollen, Joris H. Hageman, Hugo J. G. Snippert.

**Writing – review & editing:** Yannik Bollen, Joris H. Hageman, Lucca L. M. Derks, Julian R. Buissant des Amorie, Ruben van Boxtel, Hugo J. G. Snippert.

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
