## [Editor Report · Decision Letter 0]

18 May 2021

Dear Dr Snippert, 

Thank you for submitting your manuscript entitled "Efficient fluorescent gene tagging in human organoids without double-strand DNA cleavage" for consideration as a Methods and Resources article by PLOS Biology. Please accept my apologies for the delay in getting back to you as we consulted with an academic editor about your submission. 

Your manuscript has now been evaluated by the PLOS Biology editorial staff as well as by an academic editor with relevant expertise and I am writing to let you know that we would like to send your submission out for external peer review.

Please re-submit your manuscript within two working days, i.e. by May 20 2021 11:59PM.

Kind regards,

Richard

Richard Hodge, PhD

Associate Editor, PLOS Biology

rhodge@plos.org

PLOS

---

## [Decision Letter · Decision Letter 1]

17 Jun 2021

Dear Dr Snippert,

Thank you very much for submitting your manuscript "Efficient fluorescent gene tagging in human organoids without double-strand DNA cleavage" for consideration as a Methods and Resources paper at PLOS Biology. I'm handling your paper temporarily while my colleague Richard Hodge is out of the office. Your manuscript has been evaluated by the PLOS Biology editors, an Academic Editor with relevant expertise, and by three independent reviewers.

The reviews are attached below. You'll see that the reviewers find your manuscript interesting and of potential use to the field, but note that the methodology is not yet fully validated. Specifically, Reviewer #1 and #2 raise overlapping concerns with the characterization of the targeting efficiency and ask that this is calculated in the primary organoids. Reviewer #2 also asks that additional biological replicates are provided for the colonic organoids. In addition, Reviewer #3 asks for additional sequencing validation data to address potential on- and off-target mutations that could occur using the strategy. Please address all of the reviewer's concerns.

In light of the reviews (below), we will not be able to accept the current version of the manuscript, but we would welcome re-submission of a much-revised version that takes into account the reviewers' comments. We cannot make any decision about publication until we have seen the revised manuscript and your response to the reviewers' comments. Your revised manuscript is also likely to be sent for further evaluation by the reviewers.

We expect to receive your revised manuscript within 3 months. 

**IMPORTANT - SUBMITTING YOUR REVISION**

*Re-submission Checklist*

*Published Peer Review*

*PLOS Data Policy*

*Blot and Gel Data Policy*

Sincerely,

Roli Roberts

Roland G Roberts PhD

Senior Editor

PLOS Biology

rroberts@plos.org

on behalf of

Richard Hodge

Associate Editor

PLOS Biology

rhodge@plos.org

REVIEWERS' COMMENTS:

Reviewer #1:

In this manuscript, Bollen et al reported a double-strand DNA-free system to allow seamless one-step targeting vector assembly to support both N- and C-terminal fluorescent knock-ins. Three genomic loci were simultaneously modified in a single round of targeting in colon cancer organoids. Similar approaches were applied on healthy colon organoids. This can be a potential very useful tool for organoid field. However, some concerns need to be addressed before considered for publication.

Major concerns:

1. The targeting efficiency needs to be better characterized. Based on Figure 3b. All cells in one organoid express the knock-in proteins. Is this true for all organoids? If yes, what is the percentage of organoids that are triple positive, what is the percentage of organoids that express two fluorescent proteins, what is the percentage of organoids that express mScarlet, mTurquoise2, or Neongreen?

2. Is there any difference of targeting efficiency between normal and cancer organoids? The targeting efficiency should be quantified for both normal and cancer organoids as Question 1.

3. Figure S2C, sequencing of isolated clones should be performed to validate the heterozygous KI .

Minor concerns:

1. Scale bars are missing for most figure panels, which should be added and described in the legends.

2. Gating control should be added for all flow cytometry analysis.

3. Statistical analysis is missing in Figure 1e.

Reviewer #2:

[identifies herself as Joo-Hyeon Lee]

Summary 

Efficient CRISPR based gene targeting and generation of knock-ins have been challenging for 3D organoids derived from primary tissue. This manuscript describes a method to generate fluorescent reporter organoid lines by introducing homology directed repair (HDR) using Cas9 D10A nickase, which instead of creating a double stranded break (DSB) nicks only the allele targeted by the gRNA. The protocol would be beneficial to the field for efficient generation of human organoid reporter lines enabling the long-term tracing of cells. The authors showed the way to knock-in fluorescent proteins into multiple locus of the same cells, and further generate homologues lines by sequential targeting of both gene alleles. The classical antibiotics-based selection strategy could also be used in conjunction with florescence. These various plasmids used for fluorescent knock-in are also available with different fluorescent proteins to the community, proving that this protocol can be broadly useful. 

General Comments

Originality and significance: The strategy of in-trans paired nicking using modified Cas9 nicking variant has been reported in 2D culture systems such as HEK293T cells, as cited in the manuscript (Chen et al., 2017). This protocol adapted the in-trans paired nicking strategy to generate florescence-based selection in primary tissue-derived organoids, which can also be combined with antibiotics-based selection, allowing the efficient genetic engineering of primary human organoid culture models. 

Data and methodology: A major concern in this manuscript is that there is no description of biological replicates and validations in Fig. 3 and 4. For this system to be widely applicable, the authors should test multiple 'colonic' organoids and provide the detailed information of variation and efficiency of this strategy across biological repeats. 

Conclusions: The authors used HEK293T cells to calculate the editing efficiency of their strategy in Fig. 1. Given that this manuscript focuses on 3D human primary organoids, the editing efficiency of in-trans paired/cleavage method should be provided in human primary organoid culture systems with appropriate statistics.

Suggested improvements: 

1. Fig. 1: As stated above, it is crucial to provide the editing efficiency and the determination of homology arm length in human primary organoids, instead of HEK293T cells. Further, in Fig. 1b, did the authors observe the off-target effects with WT Cas9? TIDE analysis with sanger sequencing data would be useful to determine off-target effects in both WT Cas9 and nickase Cas9. In Fig. 1c, homozygous knock-in by FACS analysis must be supported by locus sequencing data either in the figure or as supplementary dataset.

2. Fig. 3: The locus that were targeted are actively transcribing/constitutively active (LMNA, HIST1H2BC, CDH1…). Can inactive locus be targeted with similar efficiency? For wider application of this protocol, both actively transcribing and inactive genomic regions must be targeted. Further, can an inactive locus in the colonic organoids targeted with a fluorescent reporter be turned-on by in vitro culture modification? OR can a florescent knock-in into an active gene be turned-off transiently? This will determine how well the florescence reports the transcriptional activity of the gene. 

3. How does the size of knock-in construct influence on the efficiency of knock-in? Is the efficiency of knock-in at CDH1 locus similar to "mScarlet" (Fig. 3) and "mScarlet-P2A-Blasticidin" (Sup Fig. 1)? How much does insert size influence on the efficiency of knock-in with this strategy in this manuscript?

4. Fig. 3 and Sup Fig. 1: How many biological repeats have been conducted for this strategy? How much efficiency variations have been observed? 

5. Have the authors tested whether organoid lines generated by the multiplex fluorescent gene tagging strategy can be stably maintained over multiple passages? Can organoid lines with 3x fluorescent reporters be frozen (liqN2), recovered and subculture for long-term tracing? It would be informative to provide these further validations for wider utility. 

6. It would be informative to provide the sanger sequencing results for the targeted locus for Fig. 3, Fig. 4, and Sup. Fig. 1.

Clarity and context: The manuscript has been well written with a logical flow and a clear abstract, introduction, and conclusion.

Reviewer #3:

[identifies himself as Andrew Bassett]

Summary

Bollen et al. demonstrate that large knock-ins in human organoids can be achieved with good efficiency and precision through use of in-trans paired nicking (nicking of both the genomic locus and targeting vector). The combination of a nicked targeting vector and nickase enzyme shows comparable overall rates of editing compared with WT Cas9, highlighting the potential of generating knock-ins without inducing a double-stranded break. The potential of this technique is further demonstrated through the generation of a triple KI in human colon organoids and the integration of different fluorescent markers on each allele at a single locus, the latter of which establishes a method for the study of allele-specific expression dynamics. Finally, to address the lengthy cloning process associated with generating targeting vectors with homology arms, the authors demonstrate a one-step targeting vector assembly strategy utilising Golden-gate cloning, resulting in a targeting vector with the specific upstream and downstream homology regions on either side of a fluorescent tag. The targeting vector backbone variants developed during this work are available through Addgene. 

I find the paper to be well written, clear and easy to understand, and the supplementary files provide a clear overview of the strategy and combined with the availability of vectors would enable others to use this methodology. Experiments were well thought out and performed, and do show the versatility and precision of in-trans paired nicking over conventional CRISPR-mediated knock-in methods. 

I would recommend acceptance in the methods section of PlOS Biology once experiments to address on and off target mutations and integrations listed below have been addressed, and the minor comments have been corrected. 

Major Issues

1. As referenced in this manuscript, the technique of in-trans paired nicking has already been established in other systems such as human cells. Thus, the principal novelty of this study is the application of this technique to human organoids 

2. Integrity of non-targeted allele. One of the major benefits of the in trans paired nicking system is that the non-targeted allele should not accumulate indel mutations, allowing for instance multiple rounds of targeting without sequence validation and cloning in-between and meaning that e.g. "…sequence verification of individually picked clones is no longer required since all knock-in cells can immediately be pooled." However, the authors do not directly demonstrate this precision. They should perform experiments to address the frequency of mutations at the non-targeted allele for instance by high throughput sequencing of the WT alleles after a targeting experiment in comparison to the DSB-mediated methodologies. 

3. On target precision. Similarly, it has been shown that DSB-mediated genome engineering can introduce undesired and unexpected mutations on target such as long deletions (Kosicki et al. NatBiotech 2018, "Repair of double-strand breaks induced by CRISPR-Cas9 leads to large deletions and complex rearrangements"), and tandem integrations of multiple copies of the transgene, etc (Canaj et al. BioRxIV 2019 "Deep profiling reveals substantial heterogeneity of integration outcomes in CRISPR knock-in experiments"). It would be important to assess the frequency of these on target mutations with the different strategies. This could be done by targeted long read sequencing (e.g. ONT), or long-range PCR and Sanger sequencing.

4. Off-target mutation. Another advantage of using Cas9 nickase is the reduction of off-target mutations. I feel that this is not addressed experimentally and should be done through sequencing potential off-target sites (in comparison to nucleases). A key reference that should be included on this topic is Chen et al. NAR 2020 "Expanding the editable genome and CRISPR-Cas9 versatility using DNA cutting-free gene targeting based on in trans paired nicking." 

5. Off-target integration. dsDNA molecules, especially when linearized, are able to integrate randomly (as well as at CRISPR off-target sites) in the genome. The authors should address the frequency at which this occurs by performing analysis of random integration of transgenes (e.g. by ddPCR copy number counting or inverse PCR).

Minor Issues

1. "By contrast, flanking homology arms with Cas9 target sites to facilitate in-trans paired cleavage saturates the homology demand at around ~600bp [10]" I'm not sure the paper they cite really demonstrates this difference, since it seems locus dependent (for instance, Fig 2C is quite similar between the paired cleavage and circular donors). They should adapt the text accordingly

2. The authors should cite additional papers describing the in trans paired nicking approach such as (but not restricted to) the following:

a. Rees et al. 2019 NatComms "Expanding the editable genome and CRISPR-Cas9 versatility using DNA cutting-free gene targeting based on in trans paired nicking " that uses a Cas9 nickase fusion to Rad51 to improve editing outcomes. 

b. Hyodo et al. 2020 CellRep "Tandem Paired Nicking Promotes Precise Genome Editing with Scarce Interference by p53"

3. Figure 1 - 5' and 3' annotation on DNA incorrect. 

4. Supplementary Figure 1 - Incorrect spelling of 'triple' in diagram title.

---

## [Decision Letter · Decision Letter 2]

7 Dec 2021

Dear Dr Snippert,

Thank you for submitting your revised Methods and Resources article entitled "Efficient and error-free fluorescent gene tagging in human organoids without double-strand DNA cleavage" for publication in PLOS Biology. I have now obtained advice from two of the original reviewers and have discussed their comments with the Academic Editor. 

As you can see, the reviewers appreciated the substantial amount of additional data included in the revised manuscript to address their comments. Based on the reviews, we will probably accept this manuscript for publication. Please make sure to address the following data and other policy-related requests that I have provided below:

(A) We note that flow cytometry data is presented in Figures 3B, 4A, S1 and S7B. We ask that you please provide the raw FCS files of these data and recommend depositing the files in the FlowRepository database due to the potential size of the files (https://flowrepository.org/). If you do use the FlowRepository, please ensure that the files are publicly available at this stage and please provide the accession number/URL for the deposition.

(B) Please deposit the raw Sanger and TIDE sequencing data (Figure S3-S5, S6 and S8) in a public database, such as the GEO. As before, please ensure that the data is made publicly available at this stage and that you provide the accession number/URL for the deposition.

(C) Please also ensure that each of the relevant figure legends in your manuscript include information on *WHERE THE UNDERLYING DATA CAN BE FOUND*, and ensure your supplemental data file/s has a legend.

(D) Please ensure that your Data Statement in the submission system accurately describes where your data can be found and is in final format, as it will be published as written there. This includes referencing where the underlying data can be found in S5_Table, as well as providing the accession numbers for the data deposited in public databases (e.g. FlowRepository and GEO).

We expect to receive your revised manuscript within two weeks. 

*Published Peer Review History*

*Early Version*

Sincerely,

Richard

Richard Hodge, PhD

Associate Editor, PLOS Biology

rhodge@plos.org

Reviewer remarks:

Reviewer #2: The revised manuscript represents a significant improvement over the initial submission. I have no further comment, and appreciate the authors' effort to address the raised questions and concerns. 

Reviewer #3 (Andrew Bassett, signs his review): The authors have substantially revised their manuscript, and it now addresses most of my concerns through incorporation of additional sequence analysis of on and off-target mutagenesis, and whole genome characterisation. The changes made have substantially improved the manuscript, which demonstrates application of this methodology to organoid systems, and also provides a toolkit of reagents for others to use to employ this. I am happy for it to be published in Plos Biology.

---

## [Editor Report · Decision Letter 3]

5 Jan 2022

Dear Dr Snippert,

On behalf of my colleagues and the Academic Editor, Madeline Lancaster, I am pleased to say that we can in principle accept your Methods and Resources "Efficient and error-free fluorescent gene tagging in human organoids without double-strand DNA cleavage" for publication in PLOS Biology, provided you address any remaining formatting and reporting issues. These will be detailed in an email that will follow this letter and that you will usually receive within 2-3 business days, during which time no action is required from you. Please note that we will not be able to formally accept your manuscript and schedule it for publication until you have any requested changes.

PRESS

Sincerely,

Richard

Richard Hodge, PhD

Associate Editor, PLOS Biology

rhodge@plos.org

PLOS
